# Neighborhood Walkability in Relation to Knee and Low Back Pain in Older People: A Multilevel Cross-Sectional Study from the JAGES

**DOI:** 10.3390/ijerph16234598

**Published:** 2019-11-20

**Authors:** Daichi Okabe, Taishi Tsuji, Masamichi Hanazato, Yasuhiro Miyaguni, Nao Asada, Katsunori Kondo

**Affiliations:** 1Advanced Preventive Medical Sciences, Graduate School of Medical and Pharmaceutical Sciences, Chiba University, 1-8-1 Inohana, Chuo-ku, Chiba-shi, Chiba 260-8670, Japan; nao.asadacar@gmail.com; 2Center for Preventive Medical Sciences, Chiba University, 1-8-1 Inohana, Chuo-ku, Chiba-shi, Chiba 260-8670, Japan; tsuji.t@chiba-u.jp (T.T.); hanazato@chiba-u.jp (M.H.); kkondo@chiba-u.jp (K.K.); 3Institute for Health Economics and Policy, 1-5-11 Nishi-Shimbashi, Minato-ku, Tokyo 105-0003, Japan; y.miyaguni@ncgg.go.jp; 4Center for Gerontology and Social Science, National Center for Geriatrics and Gerontology, 7-430 Morioka-cho, Obu City, Aichi 474-8511, Japan

**Keywords:** neighborhood walkability, built environment, musculoskeletal pain, knee pain, low back pain, older people, multilevel analysis

## Abstract

Few studies have focused on a relationship between the built environment and musculoskeletal pain. This study aimed to investigate an association between neighborhood walkability and knee and low back pain in older people. Data were derived from the Japan Gerontological Evaluation Study (JAGES) 2013, a population-based study of independently living people ≥65 years old. A cross-sectional multilevel analysis was performed, of 22,892 participants in 792 neighborhoods. Neighborhood walkability was assessed by residents’ perceptions and population density. Dependent variables were knee and low back pain restricting daily activities within the past year. The prevalence of knee pain was 26.2% and of low back pain 29.3%. After adjusting for sociodemographic covariates, the prevalence ratio (PR) of knee and low back pain was significantly lower in neighborhoods with better access to parks and sidewalks, good access to fresh food stores, and higher population densities. After additionally adjusting for population density, easier walking in neighborhoods without slopes or stairs was significantly inversely correlated with knee pain (PR 0.91, 95% confidence interval 0.85–0.99). Neighborhoods with walkability enhanced by good access to parks and sidewalks and fresh food stores, easy walking without slopes or stairs, and high population densities, had lower prevalences of knee and low back pain among older people. Further studies should examine environmental determinants of pain.

## 1. Introduction

Musculoskeletal diseases, including osteoarthritis (OA), are major public health problems. Between one in three and one in five people live with painful musculoskeletal conditions, making these diseases the second highest contributor to global disability. Low back pain alone is the leading cause of disability worldwide [1]. A strong relationship exists between musculoskeletal pain and a reduced capacity to engage in physical activity. This often results in functional decline, frailty, reduced quality of life, and loss of independence [2]. The prevalence and impact of musculoskeletal diseases are particularly high in older people. While OA may be treated surgically when severe, it is now considered amenable to prevention and treatment in the early stages [3]. For example, weight loss for obesity, prevention of injury, and exercise have all been shown to be effective in reducing knee and lower back pains [4,5] Although strong evidence supports the benefits of regular exercise, physical inactivity remains highly prevalent worldwide [6]. In fact, the number of daily steps people take in Japan is decreasing year by year, despite the fact that walking, the most frequent type of exercise, is recommended by national health policy [7,8]. For many, however, it is difficult to get regular exercise, and there are limitations to the effects of policy pronouncements at the individual level where a number of other factors are in play.

One of these factors, the built environment, has been found to exert a noticeable influence on health [9,10,11]. The World Health Organization recommends improving the built environment as a way to promote healthy aging [12]. The built environment is related to physical activity [13,14], most notably in terms of neighborhood walkability [15,16]. Neighborhood walkability is a measure of how friendly the residential built environment is to walk in. It is generally expressed as a composite index of population density, land-use diversity, and pedestrian-friendly design [17]. Neighborhood walkability has been shown to be related to time spent walking [18], physical activity [15], obesity [19], and depression [20]. These are all factors which are also well known to be associated, in one way or another, with musculoskeletal pain.

However, few studies have investigated an association between the built environment and musculoskeletal pain. If neighborhood walkability is associated in some way with musculoskeletal pain, it would become clear that not only individual factors but environmental factors can be addressed in policies designed to prevent musculoskeletal pain. Therefore, we aimed to examine whether neighborhood walkability is related to knee and low back pain, focusing on older people in Japan.

## 2. Methods

### 2.1. Study Design and Participants

The present study is based on the Japan Gerontological Evaluation Study (JAGES), an ongoing population-based cohort study in Japan [21]. In 2013, self-reported questionnaires were mailed to 193,694 community-dwelling, independently-living individuals aged 65 years or older, of whom 137,736 responded to the survey (response rate, 71.1%). Participants with missing values for ID, age, or sex (*n* = 7996); who needed assistance in activities of daily living (*n* = 4247); or people living in communities with less than 30 respondents (*n* = 2108) were excluded from the analysis. A total of 123,385 participants’ responses from 792 communities were used to evaluate neighborhood walkability. About one-fifth of the total participants (*n* = 24,806) was randomly selected, including some from each of the 792 communities, to complete a survey module enquiring about pain. The module was a planned part of the JAGES. Because long-term exposure to neighborhood walkability was considered to be beneficial, we excluded residents who had lived in their neighborhood for 3 years or less (*n =* 732). Responses were also excluded if data on knee and low back pain was missing (*n =* 1182). This left responses from 22,892 participants that were included in the subsequent analysis (Figure 1). Our research protocol and informed consent method were approved by the Ethics Committee of Nihon Fukushi University (number 13–14).

### 2.2. Outcome Variables

Data on the presence of knee and low back pain within the last year were collected in the survey by asking the following two questions. “In the past year, have you had knee pain that restricts your daily activities? In the past year, have you had low back pain that restricts your daily activities?” A response of “yes” was defined as the presence of pain.

### 2.3. Neighborhood Walkability

Many studies have established the predictive value of residents’ perceptions as a measure of neighborhood walkability [22,23]. Previously studied relationships include those between access to parks and body mass index (BMI) [24], food environment and mortality rate [25], and walking up slopes and diabetes control [26]. While some studies demonstrate that objective measures affect various health outcomes [19], two studies reported that subjective walkability rather than objective geographic information system-based data was associated with health outcomes [25,27]. Subjective walkability has the advantage of easily grasping the actual situation; for example, it can change depending on factors such as the size, number, and design during the evaluation of parks and sidewalks. Moreover, there are few studies on walkability in Japan, and the validity of objective indicators has not been sufficiently verified. Therefore, we used subjectively assessed walkability as an explanatory variable.

We evaluated neighborhood walkability by asking about access to parks and sidewalks, access to fresh food stores, and easy walking without slopes or stairs. Three questions were posed about the neighborhood within 1 km of the participant’s house. “How do you feel about access to parks and sidewalks when walking? How many stores or facilities selling fresh fruit and vegetables are located near you? How do you feel about easy walking without slopes or stairs?” Responses were given on a four-point Likert scale, with 1 = none, 2 = a few, 3 = some, and 4 = many. The average of the points in each neighborhood was used to compare each walkability variable, resulting in a minimum of 1 and a maximum of 4 continuous points. To assess neighborhood walkability, we used the data derived from all 123,385 participants rather than only the smaller subset (*n =* 22,892) of individuals who responded to the questions about knee and low back pain.

We also used population density as a variable because it is one of the main factors associated with neighborhood walkability, as it includes factors such as land-use mix, access to public transport, and number of walkable destinations [17,28]. The population density of each of the 792 communities included for analysis was calculated using the 2010 census and Land Utilization Tertiary Mesh Data (as of 2010) of the National Land Numerical Information from the Ministry of Land, Infrastructure, Transport, and Tourism in Japan based on the 1:25,000 Topographic Map of Japan [29]. These calculations excluded undeveloped areas (e.g., rivers, lakes, forest, and wasteland). Quartiles of population density (persons/km^2^) were used for analysis.

### 2.4. Covariates

For individual covariates, sociodemographic data, behavior, and health status were used. Sociodemographic covariates included sex, age (65–69, 70–74, 75–79, 80–84, and ≥85 years old), educational background (<10, 10–12, and ≥13 years), equivalent annual income (<2, 2–3.9, and ≥4 million yen per year), and past occupation (white-collar workers, blue-collar workers, primary industry workers, or never worked before) [30,31]. Primary industry workers in agriculture, forestry, and fisheries were considered separately from other blue-collar workers because those three occupations are known to be strongly associated with musculoskeletal pain [32]. Behaviors and health status covariates assessed included time spent walking (<30, 30–59, and ≥60 min a day), frequency of physical activity (<2, 2–3, and ≥4 times a week), driving status (driving a car by themselves or not) [33], BMI (<18.5, 18.5–24.9, and ≥25 kg/m^2^), and depression (none, mild, severe) [34]. Physical activity referred to medium intensity exercise, such as walking quickly, dancing, and golf [35]. Participants were classified in three groups based on scores from the Japanese version of the Geriatric Depression Scale-15 [36,37]: not depressed (<5), mildly depressed (5–9), or severely depressed (≥10) [38,39]. Missing data was counted and listed as missing.

### 2.5. Statistical Analysis

We first calculated the association between each neighborhood walkability factor and knee or low back pain using Pearson’s correlation coefficient. Multilevel Poisson regression models were then analyzed to investigate the association between neighborhood walkability and pain. An initial model was specified to assess the crude association between neighborhood walkability and knee or low back pain. This was then adjusted in Model 1 using sex, age, equivalent annual income, educational background, and past occupation as individual confounders to evaluate the influence of sociodemographic factors. Model 2 was additionally adjusted for walking time, physical activity, driving status, BMI, and depressive symptoms as potential confounders. As population density strongly affects various aspects and is easy to correlate with other walkability [28,40]. For example, in order to clarify that it is not just the influence of population density, we additionally adjusted the population density in Model 3. Using Appendix A, we identified whether covariates affected the outcomes. Stata 14.0 (StataCorp LP, College Station, TX, USA) was used, and prevalence ratios (PR) and 95% confidence intervals (CI) were calculated from the regression models. The significance level was set at 0.05. Participants with missing covariate data were still included in the analysis.

## 3. Results

The prevalence of knee pain and low back pain was 26.2% (*n =* 6257) and 29.3% (*n =* 6989), respectively (Table 1). The largest proportion by age was 70 to 74 years old (30.3%), followed by those 65 to 69 years old (28.0%). Approximately two-thirds of the participants had normal BMIs and no depression. More than a third (38.7%) walked >60 min; another third (35.2%) walked 30 to 59 min; and 23.9% walked <30 min. About half drove a car.

The means for the three subjective neighborhood walkability factors ranged from 2.56 to 2.97 (Table 2). The mean population density was 6543 persons/km^2^ (22–31,565 persons/km^2^). Reports by neighborhood of knee pain ranged from 15.6% to 51.4%, and of low back pain, from 13.6% to 51.4%. The Pearson correlations between neighborhood walkability factors were all significant. The correlations were relatively high between access to parks and sidewalks and access to fresh food stores; access to parks and sidewalks and population density; and access to fresh food stores and population density (0.44 to 0.59). There were significant negative correlations between knee pain and access to parks and sidewalks (−0.21); knee pain and population density (−0.33); and low back pain and population density (−0.17).

In the Crude regression model, knee pain was significantly less prevalent with access to parks and sidewalks, access to fresh food stores, and a high population density (Table 3). After adjustment for sociodemographic confounders (Model 1) and behavior and activity covariates (Model 2), all three walkability factors remained statistically significant. After adjusting for population density in Model 3, the only statistically significant factor associated with less knee pain was ease of walking without slopes or stairs (PR = 0.91, 95% CI = 0.85–0.99).

For low back pain, the initial results were similar to those with knee pain (Table 4). However, with Models 1 and 2, only access to fresh food stores and population density remained significantly associated with less low back pain. After adjusting for population density, ease walking without slopes or stairs fell just short being statistically significant.

## 4. Discussion

In a large and diverse, population-based sample, we found that subjectively perceived neighborhood walkability was associated with a lower prevalence of knee and low back pain. This relationship remained after adjusting for sociodemographic variables (Model 1). Although we adjusted for walking time, physical activity, driving status, BMI, and depressive symptoms as potential mediators, the association remained similar (Model 2). Even after adjusting for population density to eliminate that as a factor, one factor contributing to better walkability—ease of walking without slopes or stairs—was significantly negatively associated with knee pain (Model 3). To our knowledge, this is the first study indicating that features of the built environment may be correlated with the prevalence of musculoskeletal pain in a large-scale survey of older adults.

Earlier studies of neighborhood walkability indicated a negative association with obesity [19], which is a risk factor for knee and low back pain [3,41]. A population-based study of 9046 adults in Japan reported that living in a rural area was associated with a high prevalence of knee pain and low back pain [42]. However, that study did not adjust for occupation. The jobs of primary industry workers tend to place a heavy burden on the knee and low back, and many of these individuals live in rural areas. In our study, after adjusting for past occupation, we found that higher population density, access to parks and sidewalks and fresh food stores, and easy walking without slopes or stairs were related to lower prevalences of knee pain and low back pain.

The sociodemographic factors we assessed are considered key not only in regard to physical activity [43] and obesity [44] but to knee and low back pain, as we found relatively large changes in the PRs from the Crude Model to Model 1 after adjusting for sociodemographic factors. In fact, an association between low back pain and socioeconomic status, such as educational background, past occupations, and income, has been reported [31]. A longer time spent walking, greater physical activity, a lower BMI, and the absence of depression are factors known to be negatively related to knee and low back pain. Therefore, we initially hypothesized that these factors would be potential mediators, and as shown in the Appendix A, these factors were actually related to knee pain and low back pain. However, after adjusting for these covariates in Model 2, little change was seen in our results. Therefore, walking time, physical activity, BMI, and depression were thought to largely depend on sociodemographic status, and other factors should still be considered. Social environment variables such as social capital and safety may also be involved, as the social environment has been shown to be associated with cognitive function and social participation [45,46].

As a mechanism that might mediate the relationship between neighborhood walkability and pain, social interaction and the greenness provided by parks and sidewalks have been considered. Social interaction increases for people who frequently use parks [47] and can have a positive psychosocial influence. Good access to parks and sidewalks is likely to increase exposure to greenness which has also been shown to be associated with less obesity [48]. A fresh food store may be a place people would go every day, which would therefore encourage daily walking [25] as well as meeting friends. Such access to fresh food would also support a healthy diet that can be beneficial in preventing obesity. The relationship between walking up slopes or stairs and health is controversial [35,49]. However, to the extent that such features might hinder walking and physical activity among older adults, a flatter environment might be better in terms of walkability. Higher population density can lead to more walkable destinations, a better land-use mix, and better access to public transport and healthcare services [28]. We found that, compared with knee pain, low back pain was not significantly associated with access to parks and sidewalks or easy walking without slopes or stairs in Models 1–3. A previous review indicated that low back pain was strongly influenced by awkward posture among agricultural workers [50]. It may be, therefore, that knee pain is more closely linked with walking than is low back pain.

Strengths of this study include the focus on the association between the built environment and musculoskeletal pain in a large-scale population-based study. Past research has mainly focused on individual factors vis-à-vis musculoskeletal pain. However, it is difficult to get regular exercise and maintain a desirable weight for people with and without pain. A population-based approach should also be used for investigating musculoskeletal pain, particularly when considering public policies to prevent disability or to improve the health system [21,51]. Our results will be useful in further research on environmental determinants of pain and specific population approaches such as the primordial prevention [52], which aims for a society where people live in a health-friendly place and remain healthy without additional effort because risk factors have been minimized.

Several limitations of this study should be mentioned. First, with the exception of population density, our explanatory variables were subjectively assessed. A comprehensive scale that takes into account various factors, such as walk score or MAPS Global tool, may also be useful [53,54]. In this study, we focused on subjective indicators because it was easy to comprehend the actual situation of each element; however, evaluation of both subjective and objective indicators in the future will lead to a more detailed verification of the relationship between the built environment and pain. Second, we selected certain items that seemed to be particularly influential among various factors contributing to walkability, and that have been reported to be useful in previous studies [24,25,26]. Other variables such as street connectivity and safety may warrant inclusion in similar studies [23,55]. This study did not include them because we thought the other factors were unlikely to be related to pain alone. Further research must explore which built environment elements and scales are associated with musculoskeletal pain. Third, our outcomes included both acute and chronic pain. However, knee pain in older people is mostly due to OA [56], and the relationship weakens when other causes of knee pain are included. Therefore, it can be said that the connection to neighborhood walkability is strong. Fourth, as this is a cross-sectional study, it cannot prove a causal relationship. Exercise has been shown to have a preventive and therapeutic effect on low back pain [4,57], so better neighborhood walkability could theoretically be beneficial by improving access to exercise. People without knee pain or low back pain might choose to live in areas with good walkability, but we could not evaluate that in our study because we excluded those who have lived in the same neighborhood for 3 years or less. Longitudinal studies will be needed to better examine the nature of the relationship between neighborhood walkability and the incidence of musculoskeletal pain. Finally, although there is a high generalizability in Japan, it is difficult to generalize these results to other countries with greatly differing environments and cultures, such as those in Europe and America. In the future, aiming at the realization of a society where pain is naturally prevented, research should be conducted on whether improvement of the built environment helps reduce the prevalence of musculoskeletal pain in various regions.

## 5. Conclusions

Good neighborhood walkability with access to parks and sidewalks and fresh food stores, easy walking without slopes or stairs, and high population density were associated with a lower prevalence of knee and low back pain among older people, as demonstrated in this large-scale, population-based, multilevel analysis. Further studies should examine not only individual factors but also environmental determinants of pain.

## Figures and Tables

**Figure 1 ijerph-16-04598-f001:**
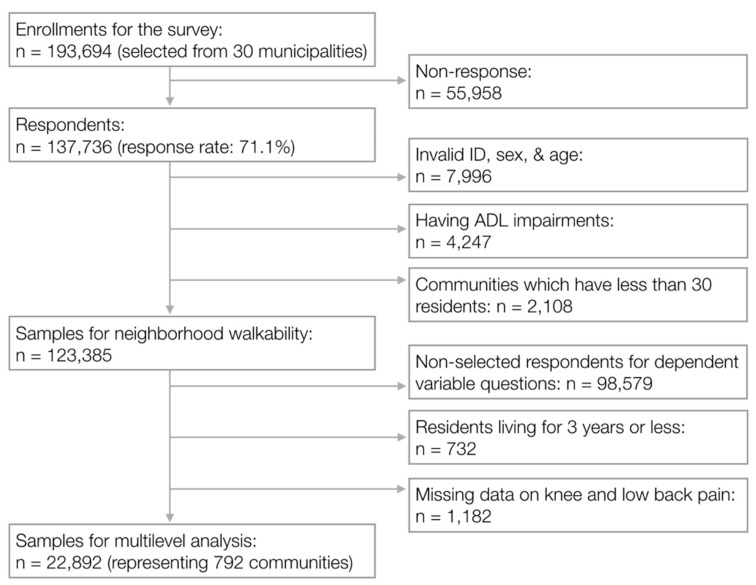
Flow of participant selection for the study of neighborhood walkability and musculoskeletal pain (*n =* 22,892). ID, identification; ADL, activities of daily living.

**Table 1 ijerph-16-04598-t001:** Characteristics of older Japanese adults surveyed in JAGES 2013 with regard to neighborhood walkability (*n =* 22,892).

Characteristics		*n*	%
Sex			
	Male	11,114	46.5
	Female	12,775	53.5
Age, years			
	65–69	6690	28
	70–74	7231	30.3
	75–79	5330	22.3
	80–84	3147	13.2
	85+	1491	6.2
Educational background, years			
	13+	4713	19.7
	10–12	8819	36.9
	<10	9974	41.8
	Missing	383	1.6
Equivalent annual income, yen			
	4.0+ million	2025	8.5
	2.0–3.9 million	7140	29.9
	<2.0 million	9875	41.3
	Missing	4849	20.3
Past occupation			
	White-collar worker	8481	37.1
	Blue-collar worker	9494	41.5
	Primary industry worker	1291	5.6
	Never worked	1170	5.1
	Missing	2456	10.7
Walking time, min			
	60+	9241	38.7
	30–59	8401	35.2
	<30	5704	23.9
	Missing	543	2.3
Physical activity			
	Daily	6858	28.7
	Weekly	6517	27.3
	Annually	3379	14.1
	None	4999	20.9
	Missing	2136	8.9
Driving status			
	No	10,854	45.6
	Yes	12,967	54.4
Body mass index, kg/m^2^			
	18.5–24.9	16,006	67
	<18.5	1665	7
	25+	6153	25.8
	Missing	65	0.3
Depression			
	None (GDS < 5)	14,223	62.1
	Mild (GDS of 5–9)	3603	15.7
	Severe (GDS ≥ 10)	1143	5
	Missing	3923	17.1
Knee pain			
	Yes	6314	26.2
	Missing	646	2.8
Low back pain			
	Yes	7050	29.3
	Missing	657	2.9

GDS = Geriatric Depression Scale; JAGES = Japan Gerontological Evaluation Study.

**Table 2 ijerph-16-04598-t002:** Pearson correlations between neighborhood walkability factors and pain.

	Mean	SD	Median	Minimum	Maximum	R					
						I	II	III	IV	V	VI
(i) Access to parks and sidewalks (score)	2.94	0.29	2.96	1.94	3.81	1					
(ii) Access to fresh food stores (score)	2.97	0.39	3.05	1.65	3.85	0.52 *	1				
(iii) Easy walking without slopes or stairs (score)	2.56	0.38	2.61	1.44	3.29	−0.17 *	0.08 *	1			
(iv) Population density (persons/km^2^)	6543	4727	6719	22	31,565	0.44 *	0.59 *	0.24 *	1		
(v) Knee pain (%)	29	6.8	27.7	15.6	51.4	−0.21 *	−0.14	0.02	−0.33 *	1	
(vi) Low back pain (%)	32.9	6.9	32	13.6	51.4	−0.14	−0.14	−0.03	−0.17 *	0.63 *	1

For neighborhood factors (i to iv), *n =* 792, while for pain (v to vi), *n =* 148, calculated only for areas with more than 30 responses about pain. For factors i–iii, the average points on a scale from 1 to 4 (1 = none, 2 = few, 3 = some, 4 = many) were calculated for each community and then combined for analysis of each factor. * *p* < 0.05. SD = standard deviation.

**Table 3 ijerph-16-04598-t003:** Association between neighborhood walkability and knee pain by multilevel Poisson regression analysis (*n =* 22,892).

	Crude Model	Model 1 ^a^	Model 2 ^b^	Model 3 ^c^
	PR (95% CI)	PR (95% CI)	PR (95% CI)	PR (95% CI)
Access to parks and sidewalks	0.69 (0.63–0.76) *	0.84 (0.76–0.93) *	0.85 (0.77–0.94) *	0.92 (0.81–1.03)
Access to fresh food stores	0.81 (0.76–0.87) *	0.90 (0.84–0.96) *	0.90 (0.84–0.96) *	0.95 (0.87–1.03)
Easy walking without slopes or stairs	1.02 (0.94–1.10)	0.96 (0.89–1.04)	0.95 (0.88–1.02)	0.91 (0.85–0.99) *
Population density	0.91 (0.89–0.93) *	0.96 (0.94–0.98) *	0.95 (0.93–0.98) *	-

PR = prevalence ratio; 95% CI = 95% confidence interval. ^a^ Model 1 was adjusted for sex, age, equivalent annual income, educational background, and past occupation. ^b^ Model 2 was adjusted for the covariates in Model 1 plus walking time, physical activity, driving status, BMI, and depressive symptoms. ^c^ Model 3 was adjusted for the covariates in Model 2 plus population density. * *p* < 0.05.

**Table 4 ijerph-16-04598-t004:** Association between neighborhood walkability and low back pain by multilevel Poisson regression analysis (*n =* 22,892).

	Crude Model	Model 1 ^a^	Model 2 ^b^	Model 3 ^c^
	PR (95% CI)	PR (95% CI)	PR (95% CI)	PR (95% CI)
Access to parks and sidewalks	0.81 (0.74–0.89) *	0.94 (0.85–1.03)	0.96 (0.88–1.06)	1.08 (0.97–1.20)
Access to fresh food stores	0.85 (0.80–0.90) *	0.92 (0.86–0.98) *	0.92 (0.86–0.98) *	0.98 (0.91–1.06)
Easy walking without slopes or stairs	1.02 (0.95–1.09)	0.98 (0.91–1.05)	0.96 (0.89–1.03)	0.93 (0.87–1.00)
Population density	0.92 (0.91–0.94) *	0.96 (0.94–0.98) *	0.96 (0.94–0.98) *	-

PR = prevalence ratio; 95% CI = 95% confidence interval. ^a^ Model 1 was adjusted for sex, age, equivalent annual income, educational background, and past occupation. ^b^ Model 2 was adjusted for the covariates in Model 1 plus walking time, physical activity, driving status, BMI, and depressive symptoms. ^c^ Model 3 was adjusted for the covariates in Model 2 plus population density. * *p* < 0.05.

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
