# Peer review of "Neighborhood Walkability in Relation to Knee and Low Back Pain in Older People: A Multilevel Cross-Sectional Study from the JAGES"

_ijerph, 2019, doi:10.3390/ijerph16234598_

Round 1

Reviewer 1 Report

The authors conducted a cross-sectional study to examine the association between the built environment and the prevalence of knee and low back pain. They concluded that good neighborhood walkability is associated with a low prevalence of knee and low back pain among elderly people. It is interesting to focus on one of the major health issues for elderly people which can restrict a participation to physical activities in relation to the influence of urban structure. While to clarify the environmental factors, which can affect musculoskeletal pain is meaningful, there remains some concerns as described below.

The authors mentioned that they decided to use subjectively-assessed walkability scale because previous studies using objective measures could not find any associations between walkability and physical activity or mortality. Practically, many reports have demonstrated the validity of objective walkability scales such as Walk Score or MAPS Global tool, and significant correlation between the walkability assessed by such scales and various health outcomes (Am J Prev Med 2013 45(2): 202–206; Int J Environ Res Public Health 2015 12: 14898-14915; JAMA 2016 315(20): 2211-2220). It could easily arise that the participants with knee pain rated their neighborhood as a poor walkability. This could result in a study bias. The point needs to be fully discussed.

The hypothesis of the study looks vague. Did the authors hypothesize that not poor walkability (bumpy road, slopes, steps) itself but low physical activity due to poor walkability result in obesity and musculoskeletal pain? If so, how do they explain the slight difference between the Model 1 and 2? Or have the authors found any significant correlation between physical activity level and the prevalence of musculoskeletal pain?

Why did the authors include only three questions to perceived assessment of walkability? As they regarded in the paragraph about limitations, there are other variables such as street connectivity, esthetic and cleanliness, safety, traffic, and so on. If these variables were considered, the result may have been different. At least, they must mention about the reason.

The authors concluded that the improvement of the built environment may help reduce the prevalence of musculoskeletal pain in older community-dwelling adults. The expression would be too much even though the study was conducted as a cross-sectional design.

Reviewer 2 Report

This study looks into the impacts of the built environment on the elderly’s health issues. I find it extremely relevant and timely as many cities around the world are experiencing rapid ageing and urgently require policy measures to address new needs and desires of the aged. Despite these several benefits, however, I cannot recommend publication of this study at its current form.

First, I question why the authors rely on a series of surveys to collect data. Especially for neighborhood walkability, a range of indices that objectively assess the walkability levels of a given area already exists and may tell us more information about the selected neighborhoods’ walkability levels or qualities.

Second, I am unsure how the sociodemographic characteristics of survey respondents are incorporated in this study. The authors use them as covariates, but it would be more relevant to identify whether the characteristics affect outcomes.

Third, I am not convinced why the authors are establishing three models for statistical analysis. What differences are there between the models? What do they each suggest?

Fourth, the authors must discuss how the findings of this study can be exported or applied elsewhere. They should critically discuss limitations of the study and provide suggestions to those who may be interested in carrying out a similar research in a totally different context.

Lastly, I cannot stop from criticizing the very first sentence of the abstract. The authors argue that a “precise relationship” between the built environment and musculoskeletal pain is unknown; and I tackle this by questioning whether a “precise relationship” really exists. Why does the relationship have to be so precise? Social science questions are not about excavating the so-called precise knowledge but about pursuing a deeper understanding of the relationships that may be complex, unlinear, or multi-faceted.

English throughout the manuscript should be significantly improved.

Round 2

Reviewer 1 Report

The authors sincerely addressed all the concerns.

Reviewer 2 Report

I don't think my comments have been well incorporated into this revision. It is up to the editor to make the final decision.